# Towards Digital Twin Implementation in Roll-To-Roll Gravure Printed Electronics: Overlay Printing Registration Error Prediction Based on Printing Process Parameters

**DOI:** 10.3390/nano13061008

**Published:** 2023-03-10

**Authors:** Anood Shakeel, Bijendra Bishow Maskey, Sagar Shrestha, Sajjan Parajuli, Younsu Jung, Gyoujin Cho

**Affiliations:** 1Department of Intelligent Precision Healthcare Convergence, Sungkyunkwan University, Suwon-si 16419, Republic of Korea; anood.shakil@g.skku.edu (A.S.); sagar@g.skku.edu (S.S.); 2Department of Biophysics, Institute of Quantum Biophysics, Research Engineering Center for R2R Printed Flexible Computer, Sungkyunkwan University, Suwon-si 16419, Republic of Korea; bizen@g.skku.edu (B.B.M.); sajjanprj@gmail.com (S.P.); isinu7@skku.edu (Y.J.)

**Keywords:** roll-to-roll gravure printing, digital twins, thin film transistors, overlay printing registration accuracy, predictive modeling

## Abstract

Roll-to-roll gravure (R2Rg) has become highly affiliated with printed electronics in the past few years due to its high yield of printed thin-film transistor (TFT) in active matrix devices, and to its low cost. For printing TFTs with multilayer structures, achieving a high-precision in overlay printing registration accuracy (OPRA) is a key challenge to attain the high degree of TFT integration through R2Rg. To address this challenge efficiently, a digital twin paradigm was first introduced in the R2Rg system with an aim to optimize the OPRA by developing a predictive model based on typical input variables such as web tension, nip force, and printing speed in the R2Rg system. In our introductory-level digital twin, errors in the OPRA were collected with the variable parameters of web tensions, nip forces, and printing speeds from several R2Rg printing processes. Subsequently, statistical features were extracted from the input data followed by the training of a deep learning long-short term memory (LSTM) model for predicting machine directional error (MD) in the OPRA. As a result of training the LSTM model in our digital twin, its attained accuracy of prediction was 77%. Based on this result, we studied the relationship between the nip forces and printing speeds to predict the MD error in the OPRA. The results indicated a correlation between the MD error in the OPRA and the printing speed, as the MD error amplitude in the OPRA tended to decline at the higher printing speed.

## 1. Introduction

Printed electronics have a monumental impact considering their suitability for several applications where conformity and cost are critical in markets [1,2,3,4]. Roll-to-roll gravure (R2Rg) is a promising approach for mass production of inexpensive and large-area flexible electronic devices [5] such as radio frequency identification tags [6], organic solar cells [7], thin-film transistors (TFTs) in active matrix-based sensor devices [8,9,10,11], and low level-integrated logic gate-based devices [12]. However, the operating frequency and level of integration still need to be improved in these devices. Thus, printed TFTs must be miniaturized to ensure high-speed operations and millions of TFTs need to be integrated in a small area which cannot be achieved easily using R2Rg. This difficulty is caused mainly by the limitations in overlay printing registration accuracy (OPRA) of the R2Rg printing system [13,14,15,16,17]. To realize inexpensive, flexible, and large-scale printed electronic devices such as simple logic gates for wireless smart labels, digital signage, and simple processors for smart packaging applications, R2Rg printed TFTs are a key device; thus, well controlled registration accuracy for printing gate electrodes and the dielectric layer and drain-source electrode should be provided [18,19]. Through well-controlled OPRA, thousands of TFTs can be integrated into a small area to allow designed functions. Therefore, well-aligned TFTs with minimal overlay printing registration error play a significant role in achieving reliable printed devices [12].

Researchers have previously conducted several studies to optimize the OPRA in R2R printing. Choi et al. proposed a nonlinear feedback controller to modulate web tension and optimize the OPRA [20,21]. They used a modified genetic algorithm to determine optimal gain values. Lee et al. developed the mathematical models for a novel scheme that can provide a rapid response in the register control of flexible substrates [22]. The results indicated successful control of the register error achieving results of under 20 µm. Kang and Kim investigated the distortion in the OPRA using a finite element (FE) and proposed an active correction algorithm for a R2R reverse-offset [12]. The study reported convergence of the OPRA to 2.4 µm and 1.9 µm in the machine direction (MD) and transverse direction (TD), respectively. Gafurov et al. evaluated the register error factors in R2R screen printing, which included several process parameters, and proposed a combined active–passive error compensation strategy [23]. The study reported 0 ± 9 µm and 0 ± 10 µm for MD and TD, respectively. Kang et al. also proposed a mathematical model that eliminates the upstream disturbances in MD and TD registers by defining them in terms of strain in both downstream and upstream directions with a time lag [24,25]. These studies have developed systematic approaches to optimize the OPRA, focusing on novel algorithms for it. However, implementing these algorithms may have limitations due to dependence on developing mechanical components or sensor technology. In Industry 4.0, manufacturing paradigms are expanding to include smart objects [26] which have brought significant attention to the concept of a digital twin [27]. The digital twin (DT) represents a system’s virtual copies that can interact reciprocally with their physical counterparts. DT has shown great potential to replicate and analyze production systems in real time [28,29]. Hence, we explore the DT for optimizing the OPRA in the R2Rg printing process using typical parameters of web tensions, nip forces, and printing speeds. This study is the first step toward implementing the DT in the R2Rg printing process to elucidate a way of optimizing the OPRA (Figure 1). Figure 1 shows a concept map for the DT application to simulate the OPRA in the R2Rg printing system. This application aims at resolving the problems of the OPRA in the R2Rg printing process to enable the printing of multilayers for fabricating TFTs and the generation of dynamic contexts to allow us to test for various scenarios using virtual copies rather than the physical system. We synchronized an internet-of-things (IoT)-based DT with the R2Rg system. Based on this synchronization, the DT could keep track of the physical system behavior. It could track past events (i.e., analyze data from completed printing sessions), constantly monitor the present system status in real time, and provide the feedback to support decision-making by predicting future possibilities which allowed us to learn the adjustments necessary to improve our system (Figure 1). Although our goal focused on developing the tools needed for building our DT to predict output characteristics of R2Rg printed-TFTs via optimizing the OPRA in the future, in this work, we also present an introductory model to show how we can predict the OPRA based on the acquired parameters in the web tensions, the nip forces, and printing speeds while running the R2Rg printing system (Appendix A). This model can optimize the printing parameters to increase the yield of the printed TFT-based devices as we correlated the OPRA with the three basic printing parameters such as the web tensions, the nip forces, and the printing speeds.

## 2. The Overlay Printing Registration Accuracy (OPRA) Prediction in R2R Gravure Printing

### 2.1. R2R Gravure Printing Process Parameters

Several printing process parameters and web handling modules can influence the OPRA. Well-known typical parameters are the web tensions, the nip forces, and the printing speeds of the R2Rg system [20]. Figure 1a shows the schematic diagram of the two-unit-R2R printing system installed at Sungkyunkwan University and indicates the various parameters of the web tensions at the unwinder, first and second printing units, and the out-feeder, as well as the nip forces at the first and second printing units.

Figure 1b–d illustrates the control system of the OPRA and shows the camera system installed in the R2Rg system to ensure the overlay printing registration. The first camera detects the printed mark on the film from the 1st printing unit, and the second camera monitors the marker on the pattern roller installed in the 2nd unit. Based on the images from the 1st and 2nd cameras, the computer calculates the misalignment between printed markers at the 1st unit as reference and markers on the pattern roller at the 2nd unit just before printing. Subsequently, the PC system corrects the position of the pattern roller on the 2nd unit to compensate for the misalignment. The 3rd camera monitors the misalignment between the two markers that occur even after correction—this being the residual error in the machine. As shown in Figure 1c, if the printing status measured by cameras (1) and (2) and monitored by camera (3) indicates misalignments along the MD, then feedback is provided to the actuator to control the speed of the second gravure cylinder to meet the OPRA in the MD. The labels (1, 2, and 3) in Figure 1b–d refer to the 1st, 2nd and 3rd cameras, respectively.

### 2.2. The Prediction Model

We analyzed the various tensions, nip forces, and printing speeds with the OPRA so as to develop the DT and hence to predict the MD error. Additionally, considering the continuous nature of the printing process, we formulated our problem as a time series forecasting model as shown in the prediction model flowchart in Figure 2. In this model, we extracted statistical features from the tension and nip force input data, such as mean, standard deviation, minimum, and maximum values over one second, followed by splitting the dataset into sequences to capture the system’s behavior over time. Subsequently, we trained the forecasting model to predict MD error. Here, we used Long Short-Term Memory (LSTM), a state-of-the-art time series forecasting method, as shown in Figure 3. LSTM is a type of recurrent neural network (RNN) architecture that Hochreiter et al. [30] developed to model sequential data accurately and learn the long-term dependency of information. The LSTM is successfully used in several fields including speech recognition [31,32,33], natural language processing [34], and stock market forecasting [35]. The LSTM contains several hidden layers with special units called memory blocks. These memory blocks store the temporal network state using self-connected memory cells. The multiplicative units in memory blocks control the flow of information and are called gates, including the input gate, forget gate, and output gate. The input and output gates control the input and output flow of cell activation, respectively. The forget gate helps adaptively reset the cell’s memory by preventing the processing of non-segmented input streams. In other words, the forget gate categorizes whether specific information should be preserved or discarded. Together, all three gates control the amount of information that flows into and out of a memory cell. Figure 3a summarizes the internal mechanism of the LSTM and how these three gates operate. Based on this mechanism, mapping from an input sequence and output were computed: using the following Equations (1) to (6), we can calculate the network unit activations iteratively from *t* = 1 to T [36]:(1)$$i_{t}=Relu\left(W_{ix}x_{t}+W_{im}m_{t-1}+W_{ic}c_{t-1}+b_{i}\right)\;$$
(2)$$f_{t}=Relu\left(W_{fx}x_{t}+W_{fm}m_{t-1}+W_{fc}c_{t-1}+b_{f}\right)\;$$
(3)$$c_{t}=f_{t}.c_{t-1}+i_{t}.g\left(W_{cx}x_{t}+W_{cm}m_{t-1}+b_{t}\right)\;$$
(4)$$o_{t}=Relu\left(W_{ox}x_{t}+W_{om}m_{t-1}+W_{oc}c_{t}+b_{o}\right)\;$$
(5)$$m_{t}=o_{t}.h\left(c_{t}\right)\;$$
(6)$$y_{t}=\phi \left(W_{ym}m_{t}+b_{y}\right)\;$$

Here, the *W* term indicates the weight matrices (for example, *W_ix_* denotes the weight matrix from the input gate to the output) while b terms denote the bias vectors (e.g., *b_i_* is the bias vector of the input gate). *Relu* is the activation function (*Relu*(*x*) = max (0, *x*)). The terms *i*, *f*, *o*, and *c* respectively represent the input gate, forget gate, output gate, and cell activation vectors, shown in Figure 3a. All these terms have the same size as the cell activation vector m. Based on the activation function Relu’s value, the three gates can have a value of either 0 or a positive number. A value of 0 would indicate that no information would pass the gate, but a positive value would allow all information to pass [37]. *g* and *h* denote the cell input and output activation functions, respectively, which here are tanh. *ϕ* is the network output activation function. In this paper, the output activation function Dense was used. Based on this internal mechanism, the memory cells of LSTM networks can store information which can be read from, stored into, and written to previous cells. Thus, relevant information from previous cells are able to enter the current cell during the sequence’s processing. Since the R2Rg printing is a continuous process, in which the register error does not only depend on the system’s current state but also on its previous state, the LSTM is suitable for modeling and predicting the register error. Figure 3b shows the deep LSTM architecture adapted from [31,32,33] and used in this work. Deep LSTM is built by stacking several LSTM layers, and for our deep LSTM, we stacked two LSTM layers. Since the LSTM can only provide the internal state of the hidden units as output, a Dense layer is added. The Dense layer can change the LSTM output vector dimensions to those desired so that the model can easily interpret the input–output relationship. Deep LSTM RNNs offer another benefit over other RNNs: they can make better use of parameters by distributing them over the space through multiple layers. This was useful for us, as it meant we did not need to increase the memory size of our network [36].

The loss function was defined, in which the minimization problem was solved using the equations above to find the optimal *W* value. We used a mean square error loss function (*MSE*) defined using the equations below [38]:(7)$$MSE=\frac{1}{n}\overset{n}{\underset{t=1}{\sum }}{\left(y_{t}-{\hat{y}}_{t}\right)}^{2}$$
where $y_{t}$ is the observed output (the MD register error here), and ${\hat{y}}_{t}$ is the predicted error. In machine learning, after training the model, a metric must be chosen to evaluate its performance. Since our output is a continuous variable, we used the coefficient of determination, also known as $R^{2}$ score [39]. We can use the equation below to calculate this metric:(8)$$R^{2}=1-\frac{\Sigma_{t=1}^{n}{\left(y_{t}-{\hat{y}}_{t}\right)}^{2}}{\Sigma_{t=1}^{n}{\left(y_{t}-\bar{y}\right)}^{2}}$$
where $\bar{y}$ is the mean of the observed output values and can be calculated as follows:(9)$$\bar{y}=\frac{1}{n}\overset{n}{\underset{t=1}{\sum }}y_{t}$$

## 3. Experimental Results

### 3.1. Data Acquisition

To train the error prediction model in the MD, we obtained tension, nip force, MD OPRA, and TD OPRA data from several printing processes at the R2Rg printing system installed at Sungkyunkwan University. The data was acquired at a printing speed of 90 mm/s. Figure 4 shows the effects of the OPRA on the variation of threshold voltage (*V_th_*) in the R2R printed TFTs. To calculate the threshold voltage variation values depicted in Figure 4, eight randomly selected TFT samples were used. When printed layers are normally aligned (Figure 4a), the V_th_ mean values and variation are relatively low. On the other hand, the MD misalignment in the OPRA resulted in the larger *V_th_* mean values and variation compared with the normal and the TD misalignment samples. Therefore, in this work, we focused on optimizing the MD misalignment error in the OPRA, as this helped minimize the variation of R2Rg printed TFTs.

Figure 5 shows samples of the raw data obtained during a printing process, including the input data of nip force and tension (Figure 5a,b) with the printing speed of 90 mm/s. We observed a variation of 2 kgf and 0.5 kgf in the nip force and the web tension, respectively. Figure 5c,d shows the data obtained from the camera system for MD and TD misalignment of printed markers, respectively. Figure 5e shows the MD register errors attained from the camera data. These peaks represent the local minima and maxima in the camera data for MD by simply comparing the neighboring values, while Figure 5f shows MD errors computed by subtracting the peaks from an average offset value in the MD camera data. MD error was observed in the range of 50 to 100 µm while we theoretically expected it to be close to zero with the camera control system.

### 3.2. Prediction Model Training and Evaluation

Based on the framework described in Figure 2, we used the collected data during the printing process to train the deep LSTM model (Figure 3b). First, to prepare the training dataset, we manually extracted statistical features from each input variable for a one-second time window, including mean, standard deviation, minimum, and maximum values. Table 1 shows a sample of the computed statistical features. Second, we mapped the extracted input features and MD register error data. Subsequently, we divided the dataset into sequences of 10-s length to accurately capture the system behavior. Appendix A elaborate the mapping algorithm used to map the camera data to tensions and nip force data in order to prepare the dataset for training, testing and evaluating our model.

Figure 6a shows the training results of the prediction model. The training and validation loss converges to a constant value after epoch 150 (training iteration). Figure 6b shows the predicted and observed MD error from new printing processes. We were able to see how well the model can capture the system’s behavior. Table 2 reports the *R^2^* value when training the model at various input sequence lengths. The attained *R^2^* value represents the coefficient of correlation; a measure of accuracy for the prediction model as described in Section 2.2. At a sequence length of 10 s, we obtained the highest *R^2^* value of 77% indicating the best model performance.

## 4. Effects of the Nip Force Variation on MD Error under Various Printing Speeds

Since we had an introductory DT that could predict the system behavior, we used it to elucidate the effects of varying the nip force and the printing speed on the MD misalignment and to extract printing conditions by which we could optimize the OPRA. We collected data from the R2R gravure system without actual printing at different nip force levels and printing speeds, similar to the data obtained during printing. We used the introductory DT to acquire the MD misalignment error values. Since the experiment was carried out without actual printing, there were no observed MD error values.

Table 3 summarizes the conditions by which we collected data and ran the DT with a total of 3 × 4 = 12 combinations of nip forces and printing speeds. Figure 7a,b shows samples of the raw data of nip force collected over a five-minute interval without actual printing. In Figure 7a, we can observe that the noise frequency in the nip force changed when the nip force was at a specific level (here 6 kgf), and the printing speed varied. Similarly, at the printing speed of 90 mm/s, we noted that the noise frequency in the nip force remained constant at all levels as shown in Figure 7b. We assume that the nip force fluctuation is due to off centric rotation of the gravure cylinder (Appendix A).

Figure 8 shows the predicted error from our trained LSTM model as a function of the time domain (Figure 8a,b) and as a function of the frequency domain (Figure 8c,d) under the printing conditions depicted in Figure 7a and 7b, respectively. We observed a relative decline in the MD misalignment error at the higher printing speeds (Figure 8a). Also, MD misalignment error was predicted under the variation of nip forces between 4 to 8 kgf (Figure 8b). Figure 8c,d shows the analysis of the predicted error as the function of the frequency domain. We can observe that the predicted error noise frequency peaks varied at different speeds and increased in proportion to the increase in printing speed (from 0.07 Hz at 30 mm/s to 0.3 Hz at 120 mm/s in Figure 8c). On the other hand, we noticed that the frequency of the error signal remained constant when printing speed was constant, while the nip force changed—as shown in Figure 8d where the printing speed for all three conditions was 90 mm/s and error signal noise frequency peak remained around 0.22 Hz. One major reason which can explain these observations originates from the variation in the number of printed markers occurring per rotation while varying the printing speed, which further validated the predictability of our DT. These observations further strengthen the correlation between nip force noise frequency, printing speed, and predicted error frequency. Appendix A provides additional analysis of nip force, observed MD error, and predicted MD error as a function of time and frequency domains. These data were acquired from a real printing session and further validated our results. We calculated the frequency of noise in the nip force (Appendix A) and found it to be similar to noise frequency in the register error from actual printing. From this observation, we hypothesized that the nip force and register error will always have the same noise frequency. Hence, we manipulated the nip force noise frequency by varying the printing speeds and obtained a predicted error from our model. As hypothesized, the error frequency was similar to nip force at varying printing speeds.

## 5. Conclusions

In this paper, we presented the foundations for adapting the digital twin paradigm into the R2Rg system for printing TFTs with multilayer structures where each layer should be well optimized in the OPRA. We focused on developing the MD register error prediction model as the first step toward predicting future events in the digital twin paradigm, since the MD misalignment brought larger V_th_ variation than that from the well aligned and TD misalignment samples. Thus, we proposed using a deep Long-Short Term Memory (LSTM) neural network architecture as a time series forecasting method. The introductory DT used R2Rg printing parameters including web tension and nip force as register error predictors. We evaluated the performance of the deep LSTM at different sequence lengths and obtained the best accuracy of 77% at a 10 s input sequence length. This accuracy indicated the validity of using tension and nip force parameters in the R2Rg system as input variables of an LSTM-based DT to forecast MD registration error.

The paper explored the effects of varying nip force and printing speeds on the predicted register error using the proposed introductory DT. Evaluating the results in the time domain denoted a decline in MD error at higher printing speeds. Moreover, analyzing the predicted error in the frequency domain indicated a correlation with printing speed and nip force signal frequency. This finding further validated the DT’s performance.

In future work, we will focus on the link to feed process data from the R2Rg system into our DT in real time. We will further investigate process parameters, such as ink rheology and printed pattern quality which may affect the MD error and include them to expand our model.

## Figures and Tables

**Figure 1 nanomaterials-13-01008-f001:**
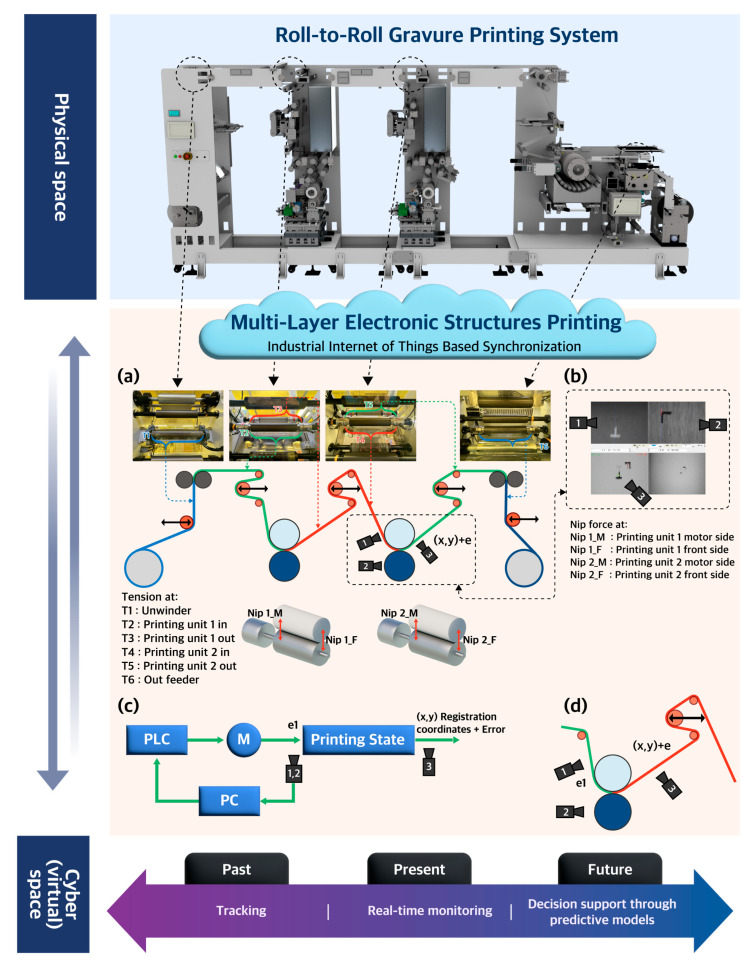
The conceptual map of introductory digital twin in roll-to-roll gravure (R2Rg)-based manufacturing system: (**a**) a schematic diagram of R2Rg system with all tension and nip force modules including tensions T1 to T6 and nip forces (NIP1_M, NIP1_F, NIP2_M, and NIP2_F); (**b**) a picture of the installed camera system used to measure and compensate the misalignments of the printed markers; (**c**) a block diagram for the OPRA control system; and (**d**) a camera system to monitor misalignments in printed markers between first (camera 1) and second (camera2) printing units.

**Figure 2 nanomaterials-13-01008-f002:**
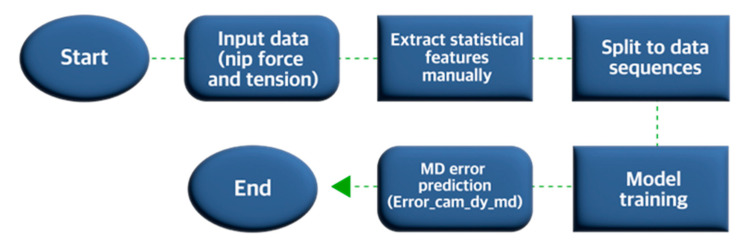
Diagram illustrating the introductory digital twin based on a predictive model.

**Figure 3 nanomaterials-13-01008-f003:**
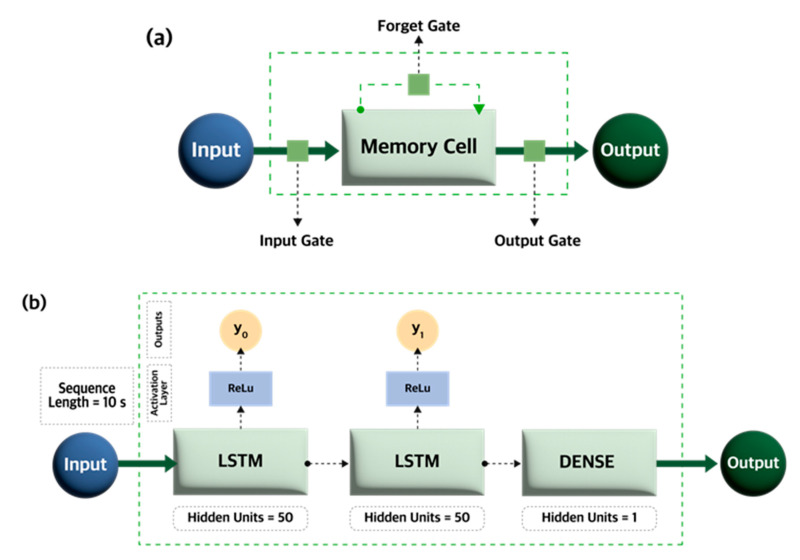
The prediction model architecture: (**a**) LSTM internal structure and (**b**) deep LSTM.

**Figure 4 nanomaterials-13-01008-f004:**
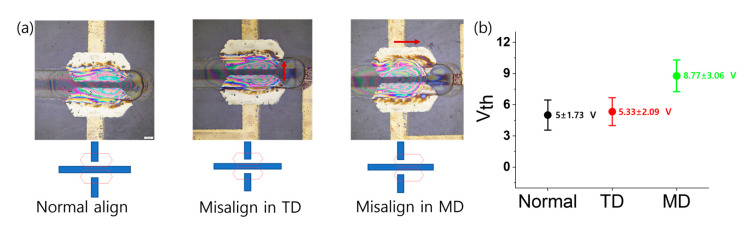
Effects of the OPRA on the variation in threshold voltage (V_th_) of the R2R printed TFTs: (**a**) optical image of TFTs showing OPRA with normal alignment, misalignment in TD and misalignment in MD; and (**b**) Vth mean value variation for TFTs with normal alignment, misalignment in TD and misalignment in MD.

**Figure 5 nanomaterials-13-01008-f005:**
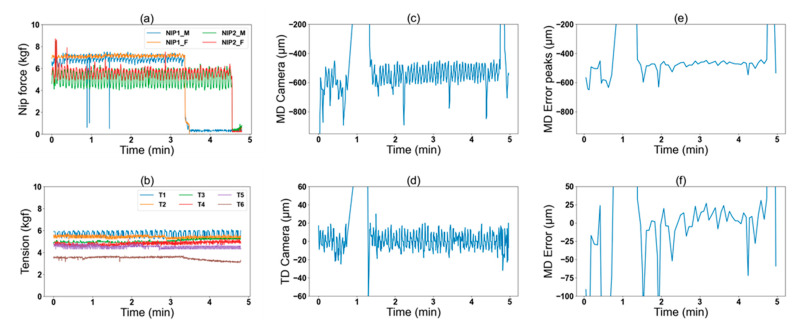
Raw data from the R2R gravure printing processes: (**a**) the nip force data from first and second printing units, (**b**) the web tension data from unwinder, first and second printing units, and out-feeder, (**c**) registration errors for MD, (**d**) registration errors for TD, (**e**) peak values from the camera data, obtained by computing the local minimums and maximums in MD registration errors (shown in (**c**)), and (**f**) the MD error calculated by estimating an offset line around which the registration error peaks oscillate, and then subtracting the offset value from the peaks graph.

**Figure 6 nanomaterials-13-01008-f006:**
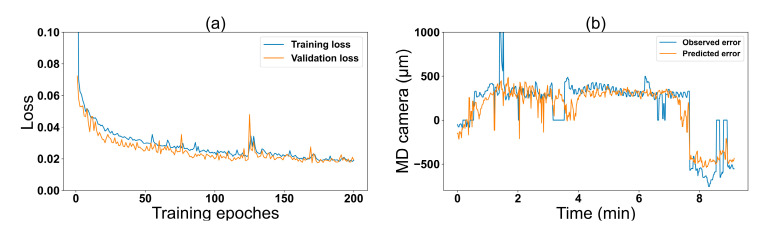
Training results: (**a**) training and validation mean square error (MSE) loss graphs for each epoch and (**b**) actual and predicted MD errors.

**Figure 7 nanomaterials-13-01008-f007:**
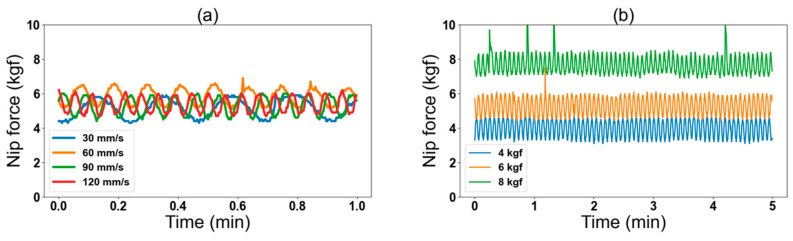
(**a**) NIP2_M data at varying printing speed and (**b**) NIP2_M data at different nip forces.

**Figure 8 nanomaterials-13-01008-f008:**
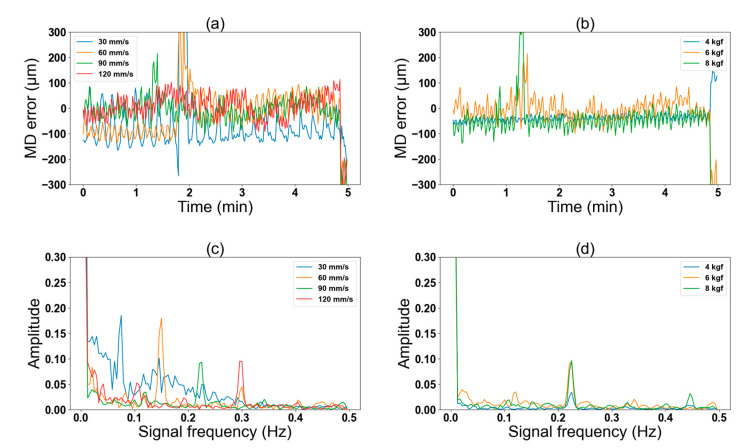
Effects of nip force variation on machine directional (MD) error by varying printing speeds: (**a**) predicted MD error at different printing speeds (set nip force = 6 kgf) and (**b**) at different nip forces (set printing speed = 90 mm/s) as a function of the time domain; (**c**) predicted MD error at different printing speeds (set nip force = 6 kgf) and (**d**) at different nip forces (set printing speed = 90 mm/s) as a function of the frequency domain.

**Table 1 nanomaterials-13-01008-t001:** Statistical features computed from the input data.

T2_Mean	T2_Std	T2_Min	T2_Max	…	NIP2_F_Min	NIP2_F_Max
4.92	0.192353841	4.6	5.1	…	5.6	6.1
4.86	0.260768096	4.6	5.2	…	6.3	7.0
4.92	0.044721361	4.9	5.0	…	7.1	7.8
4.82	0.334664011	4.5	5.3	…	6.1	7.2
…	…	…	…	…	…	…
5.12	0.083666003	5.0	5.2	…	5.5	6.9
5.14	0.089442719	5.0	5.2	…	6.0	7.4

**Table 2 nanomaterials-13-01008-t002:** *R^2^* value for different sequence lengths.

Sequence length (seconds)	4 s	6 s	8 s	10 s
*R^2^* value	62%	71%	66%	77%

**Table 3 nanomaterials-13-01008-t003:** The nip-speed experiment parameters.

Process Parameter	Units	Values
Nip force	Kgf	[4, 6, 8]
Printing speed	mm/s	[30, 60, 90, 120]

## Data Availability

Data can be obtained on request from anood.shakil@g.skku.edu. It may also be accessed from Appendix A after publication of manuscript.

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
