# Peer review of "Towards Digital Twin Implementation in Roll-To-Roll Gravure Printed Electronics: Overlay Printing Registration Error Prediction Based on Printing Process Parameters"

_nanomaterials, 2023, doi:10.3390/nano13061008_

Round 1

Reviewer 1 Report

This paper examines the improvement of registration accuracy and error
prediction in RTRg using registration monitoring with cameras
and machine learning.
The obtained results are reasonable,
and can be said to be the data that will form the basis for
improving the equipment, and will lead to future development.
Therefore, this paper has a potential for publication.

Author Response

Dear Andrijana Pavlovic, Assistant Editor:

It is our great pleasure to revise the manuscript according to the reviewers’ comments for consideration by Nanomaterials. Please refer to the indicated pages and lines of the revised manuscript to confirm the revisions.

Reviewer 1 recommended to publish our manuscript as it is. We thank the reviewer for his or her time and effort to towards reviewing our manuscript.

Reviewer 2 Report

This paper reports a detailed study on roll-to-roll process parameter control. The results and conclusion are sound. The paper covers solid progress which deserves to be published in Nanomaterials. I suggest the paper can be accepted after minor revision on English style. One minor suggestion is that it's better to add a paragraph on discussing the applications of TFTs at such printing scale. This will be helpful to practical users.

Author Response

Dear Andrijana Pavlovic, Assistant Editor:

It is our great pleasure to revise the manuscript according to the reviewers’ comments for consideration by Nanomaterials. Please refer to the indicated pages and lines of the revised manuscript to confirm the revisions.

Reviewer #2 (Remarks to the Author):

  1. Reviewer 2 recommended adding more applications regarding to TFTs. Thank you for bringing this issue to our attention. It would be indeed useful for practical users to learn about large-scale TFTs applications. To realize inexpensive, flexible, and large-scale printed electronic devices such as simple logic gates for wireless smart labels, digital signages, and simple processors for smart packaging applications, R2Rg printed thin film transistors are a key device so that well controlled registration accuracy for printing gate electrodes, dielectric layers, active layers, and drain-source electrodes should be provided. Through well-controlled overlay printing registration accuracy, thousands of TFTs can be integrated into a small area to show designed functions. Therefore, well-aligned TFTs with minimal overlay printing registration error plays a significant role in achieving reliable printed devices. We have added this part to page 1, lines 41-44, page 2, lines 45-48.

To realize inexpensive, flexible, and large-scale printed electronic devices such as simple logic gates for wireless smart labels, digital signages, and simple processors for smart packaging applications, R2Rg printed TFTs are a key device so that well controlled registration accuracy for printing gate electrodes, dielectric layer, and drain-source electrode should be provided [18] [19]. Through well- controlled OPRA, thousands of TFTs can be integrated into a small area to show designed functions. Therefore, well-aligned TFTs with minimal overlay printing registration error play a significant role in achieving reliable printed devices [12].

  1. Reviewer 2 gave general comments about editing for English style. Thank you again for the valuable comments regarding polishing the manuscript for better readability. We have carefully revised this manuscript again.

In the revised manuscript, the corrections and additional explanations with new references 

Reviewer 3 Report

In this paper, the authors studied how nip forces and printing speeds relate to predicting MD errors in the OPRA, with a focus on accuracy. The findings demonstrate that a correlation exists between the MD error in the OPRA and the printing speed. Specifically, the amplitude of the MD error at OPRA 25 decreases at higher printing speeds.

This paper needs a revision before it can be published. Please see my following comments for details.

1. How does the use of a deep LSTM architecture, consisting of stacked LSTM layers with a Dense layer, with activation functions such as Relu and tanh, and weight matrices and bias vectors for input, forget, output, and cell activation vectors, contribute to the interpretation of input-output relationships in machine learning models?

2. It is unclear why the use of Long Short-Term Memory (LSTM) in a time series forecasting model, which incorporates statistical features extracted from tension and nip force input data and captures the behavior of the printing system over time, effectively predict MD error in the printing process, and what are the specific roles and functions of the LSTM architecture's memory blocks and gates in storing and controlling the flow of information within the model?

3. How effective is the deep Long-Short Term Memory (LSTM) neural network architecture as a time series forecasting method for predicting future events in the digital twin paradigm using the R2Rg printing parameters including web tension and nip force as register error predictors?

4. Some statements in the article do not have references. For instance, when the authors listed thin-film transistors (TFTs) in the introduction, they should cite related paper and esp. reference articles (Organic Electronics, 103, 106448, (2022); Materials Advances, 4, 769-86, (2023)) in order to provide readers background information.

5. Can the proposed introductory DT predict the register error accurately by varying the nip force and printing speeds in the R2Rg system for printing TFTs with multilayer structures?

6. How does the correlation between printing speed and nip force signal frequency, analyzed as the function of frequency domain, validate the performance of the DT in predicting the register error?

Author Response

Dear Andrijana Pavlovic, Assistant Editor:

It is our great pleasure to revise the manuscript according to the reviewers’ comments for consideration by Nanomaterials. Please refer to the indicated pages and lines of the revised manuscript to confirm the revisions.

Reviewer #3 (Remarks to the Author):

  1. Reviewer 3 was curious how the use of a deep LSTM architecture, consisting of stacked LSTM layers with a Dense layer, with activation functions such as Relu and tanh, and weight matrices and bias vectors for input, forget, output, and cell activation vectors, contribute to the interpretation of input-output relationships in machine learning models. Reviewer 3 also suggested adding the explanation of why the use of LSTM in a time series forecasting model, which incorporates statistical features extracted from tension and nip force input data and captures the behavior of the printing system over time, effectively predicts MD error in the printing process, and what are the specific functions of the LSTM architecture's memory blocks and gates in controlling the flow of information within the model. Thank you for your valuable comments. Many fields, including speech recognition and stock market prediction, utilize the LSTM models. LSTM is mainly used to learn the long-term dependencies of the information. As our R2Rg printing is a continuous system in which the register error depends on the present and previous system states, LSTMs have great potential for modeling MD register errors. An LSTM unit consists of forget, input, and output gates, which control the information that can flow into and out of the memory cell and learn the useful aspects of the predictor variables, namely, web tension and nip force in this work. The forget gate categorizes the information that should be discarded or kept. The input gate updates the cells, and the output gate decides the next hidden state that can pass to the following LSTM unit. As described in equations (1, 2, and 4) in section 2.2 (page 5, lines 146-151), we can observe that the activation function Relu determines the values of the multiplicative gates input, forget, and output, which can be either “0” or a positive number. A value of “0” means no information can pass through the gate, but a positive value would indicate that all the information will pass through the gates. Each cell inside the LSTM memory block can store the information that can be read from, written to, and stored in the previous cell via the gates described above. Hence, relevant information from the previous cells can enter the current cell throughout the sequence’s processing. This mechanism implies that during the model training, it will learn both the current and past state of the web tension and nip force, which result in a misalignment error in printing. We have added this explanation to page 4, lines 134-136, 142-145, and page 5, lines 162-164, 167-173, 178-180.
  2. Reviewer 3 is also curious about how effective is the deep Long-Short Term Memory (LSTM) neural network architecture as a time series forecasting method for predicting future events in the digital twin paradigm using the R2Rg printing parameters, including web tension and nip force as register error predictors. Thank you for your comments and query. Based on our model evaluation results, it has an accuracy of 77% based on the R2 score metric calculated in Section 3.2 in Table 2 (page 8, line 254). This limited accuracy is due to the limited number of predictors. Since R2Rg printing is a complex process, the model has the potential to upgrade by introducing additional variables such as roller markers, ink transfer, and others, as highlighted in the Conclusion. Furthermore, as we focus on a few variables, the model's architecture is relatively simple (two LSTM and one Dense layer), which helps avoid overfitting. In our future work, if we incorporate new variables into our model, we may modify the structure of the model to enhance the predictive accuracy.
  3. Reviewer 3 recommends that some statements in the article do not have references. For instance, when the authors listed thin-film transistors (TFTs) in the introduction, reviewer suggested to cite related papers and esp. reference articles (Organic Electronics, 103, 106448, (2022); Materials Advances, 4, 769-86, (2023)) to provide readers background information. Thank you for pointing this out. We have tried our best to cite and provide additional references that readers will find helpful in understanding the background of our study and the methods used. We have included additional references [9] and [11] respectively. We also added references for Section 2.2 [Ref [31-35] and Ref [37] to support the additional explanations.
  1. Reviewer 3 is concerned about whether the proposed introductory DT can predict the register error accurately by varying the nip force and printing speeds in the R2Rg system for printing TFTs with multilayer structures. Thank you for your valuable comment. The introductory DT can predict the MD register error in printing each layer of TFTs with accuracy of 77% based on the input data of web tensions and nip force variables which we have explained in page 8, lines 243-250). Adding parameters in addition to web tension and nip force can further improve prediction in registration accuracy for multilayer structures.

  1. Reviewer 3 is curious how the correlation between printing speed and nip force signal frequency, analyzed as the frequency domain’s function, validates the DT’s performance in predicting the register error. Table 3 shows the experiment carried out to further validate the developed DT by running the R2Rg at different speeds and nip forces without actual printing and thus without real observed error. Based on the collected nip forces and printing speeds, MD error is predicted with the trained LSTM model. Running R2Rg machine at various printing speeds resulted in predicted MD error at different noise frequency peaks (Figure 8(c)) which is due to the faster occurrence of printed reference markers (8 markers in 1 rotation) at higher speeds. We assume that nip force fluctuation is from roller eccentricity. Thus, nip force noise frequency is proportional to printing speed and pattern roller diameter. The same noise frequency is also observed on predicted MD error depending on varying printing speed (Figure 8(c)). We calculated the frequency of noise in the nip force (Figure S2 (a) and (d)) and found it to be similar to noise frequency in the register error from actual printing. From this observation, we hypothesized that the nip force and register error will always have the same noise frequency. Hence, we manipulated the nip force noise frequency by varying the printing speeds and obtained a predicted error from our model. As hypothesized, the error frequency was similar to nip force at varying printing speeds. This result validated the reliability of our model. We added these explanations in page 9 and lines 292-298.

We calculated the frequency of noise in the nip force (Figure S2 (a) and (d)) and found it to be similar to noise frequency in the register error from actual printing. From this observation, we hypothesized that the nip force and register error will always have the same noise frequency. Hence, we manipulated the nip force noise frequency by varying the printing speeds and obtained a predicted error from our model. As hypothesized, the error frequency was similar to nip force at varying printing speeds.

In the revised manuscript, the corrections and additional explanations with new references are accentuated with red enabling track changes to facilitate the review. We hope that the responses above sufficiently address the reviewers’ comments and meet your expectations.

Please contact us at your earliest convenience if you need any further information.

Reviewer 4 Report

see the attached report

Author Response

Dear Andrijana Pavlovic, Assistant Editor:

It is our great pleasure to revise the manuscript according to the reviewers’ comments for consideration by Nanomaterials. Please refer to the indicated pages and lines of the revised manuscript to confirm the revisions.

Reviewer #4 (Remarks to the Author):

  1. On page 4, line 123, Reviewer 4 is curious that the authors introduced the Long Short-Term Memory (LSTM), a state-of-the-art time series forecasting method as the predictive model to model the sequential data in the R2R gravure printing process. The Reviewer recommends to give detailed description of the advantages of LSTM compared with the other neural network (RNN) architecture. Thank you for your valuable Deep learning tasks such as speech recognition, natural language processing, and stock market prediction utilize the LSTM model. The LSTM is used to learn the long-term dependence in the information. The internal memory and cell mechanisms set LSTMs apart from other networks. An LSTM unit consists of forget, input, and output gates, which control the information that can flow into and out of the memory cell. The forget gate categorizes the information that should be discarded or kept. The input gate updates the cells, and the output gate decides the next hidden state that can pass to the following LSTM unit. As described in equations (1, 2, and 4 in page 6, lines 142-148), we can observe that the activation function Relu determines the values of the multiplicative gates input, forget, and output, which can be either 0 or a positive number. A value of 0 means no information can pass through the gate, but a positive value would indicate that all the information will pass through the gates. Each cell inside the LSTM memory block can store the information that can be read from, written to, and stored in the previous cell via the gates described above. Hence, relevant information from previous cells can enter the current cell throughout the sequence’s processing. Deep LSTM RNNs offer many benefits over other RNNs. They can better use parameters by distributing them over the space through multiple layers. For instance, rather than increasing the memory size of a standard model by a factor of 2, one can have four layers with approximately the same number of parameters. This distribution results in inputs going through more non-linear operations per time step. We have added this explanation to page 4, lines 134-136, 142-145, and page 5, lines 162-164, 167-173, 178-180.
  2. On page 6, line 176, the authors claim the effects of the OPRA on the variation of threshold voltage (Vth) in the R2R printed TFTs, and report the Vth mean value variation for TFTs with normal alignment, misalignment in TD, and misalignment in MD in Figure 4(b), for improving the reliability of the published data, the authors need to disclose the number of samples used. Thank you for pointing that out. For calculating the variation of threshold voltage Vth in the printed TFTs, we have used randomly selected 8 samples in this study. We have added this explanation to page 6, lines 201-202. To calculate the threshold voltage variation values depicted in Figure 4, randomly selected eight TFT samples were used.

  1. Reviewer 4 recommends modifying Figure 1 to give authors a more straightforward message by adding a space between the words “things” and “based” in the middle of this Figure. Thank you for your valuable recommendations. We have modified Figure 1 while keeping the readability aspect in mind. Please refer to the modified version of the Figure in the manuscript.

In the revised manuscript, the corrections and additional explanations with new references are accentuated with red enabling track changes to facilitate the review. We hope that the responses above sufficiently address the reviewers’ comments and meet your expectations.

Please contact us at your earliest convenience if you need any further information.

Round 2

Reviewer 3 Report

Paper can be published as it is.